# "Alleged Disabilities": The Evolving Tactics of Child Protection in a Disability Rights Environment

**Hanna Björg Sigurjónsdóttir * and James Gordon Rice**

School of Social Sciences, University of Iceland, 102 Reykjavík, Iceland
* Correspondence: hbs@hi.is

**Abstract:** This contribution reports on a child protection case concerning the removal of a child from the custody of a parent with intellectual and developmental disabilities (IDD) in Iceland. Employing a mix of document analysis and interviewing, the results demonstrated two key themes forming the analysis: One is the aura of professionalism. A careful examination of the working methods reveals a continuation of the poor practices typical of the past, despite the claims made that specialised support for persons with disabilities has been tried and was not successful. The second analytical theme is alleged disabilities. This case provided evidence of a previously unseen tactic, to the best of our knowledge, by which a parent's disability status was called into question. The argument offered herein is that this was pursued to sidestep the protections afforded to disabled parents under Icelandic law in recent years. We conclude by arguing that the combination of a heighted awareness of these legal protections and a greater scrutiny as to how these cases are worked appears to have led to a series of evolving tactics that are employed against disabled parents in an enhanced disability rights environment.

**Keywords:** Iceland; disability; parents with disabilities; child protection; disability rights; custody deprivation; intellectual disabilities; CPRD; disability studies





## 1. Introduction

Mukti Jain Campion wrote over twenty-five years ago: "Disability and parenthood are words which still seem to come together only uncomfortably in our society" (Campion 1995, p. 133). Despite the apparent progress that has been made in advancing the cause of disability rights in the years since, as seen, for example, with the United Nations Convention on the Rights of Persons with Disabilities (United Nations 2006), Campion's words unfortunately continue to ring true in the present. This is particularly the case in child protection interventions concerning parents with intellectual or developmental disabilities (IDD). Campion articulates several of the common negative associations between disability and parenting, such as fears that the child will inherit the impairments of the parent; that the supposed dependency of the parent means that disabled people are the recipients of care and not care providers themselves; and that the children of disabled parents will be neglected, among others (Campion 1995, pp. 135–42). Many of the issues that Campion mentions are findings shared by other scholars working in the same period, particularly in regard to how families headed by parents with IDD fare in child protection systems (see, e.g., Booth and Booth 1994; McConnell and Llewellyn 1998, 2000; Hayman 1990). There is a consensus among scholars working in this field that there are a number of problematic issues with regard to how parents with IDD are viewed and treated by child protection systems. Our collective research experiences and findings in Iceland (Stefánsdóttir et al. 2022a, 2022b; Rice and Sigurjónsdóttir 2022, 2019; Rice et al. 2021; Sigurjónsdóttir and Rice 2017, 2018, 2020) generally mirror those of the international literature, particularly the works with which we are the most familiar from Nordic countries, North America, the UK, and Australia. This research was produced to a large extent by our colleagues in the IASSIDD

(The International Association for the Scientific Study of Intellectual and Developmental Disabilities) parenting special interest research group (SIRG). Formal support measures play a significant role in the lives of parents with IDD, particularly those without strong informal support networks (Collings et al. 2017). Additionally, where formal support is available, it is often either inadequate or poorly suited to their needs (Collings et al. 2017; MacIntyre and Stewart 2012; McConnell and Llewellyn 1998), or provided by staff who are not well trained or who implement the support in what parents perceive as a condescending, paternalistic, or even hostile manner (Starke 2010; Strnadová et al. 2017). Parenting capacity assessments play a significant role in custody deprivation cases, yet in the case of parents with IDD, there are significant limitations to these assessments that need to be considered and often are not (Spencer 2001). Furthermore, it is not always clear in custody deprivation cases whether the impairments are being diagnosed properly, and there are concerns that the parents are thus not receiving the appropriate support and are "falling through the cracks" (Lightfoot et al. 2017). In our experience, in Iceland, an additional problem in these cases is that what is claimed by child protection (and presented to the courts) to be "support" often, in practice, consists of surveillance, brief visits, and counselling conducted by individuals who do not always have professional knowledge or experience working with parents with IDD. This support is certainly not the kind of hands-on, evidence-based skill teaching that these parents require, as seen in programs such as Step-by-Step (Feldman 2004, 2020). We also repeatedly encountered poorly organised support plans in which measures were directed toward the parent, parents, or children, without much in the way of consideration for the family as a collective unit. This is also a notable feature of the international findings (Collings et al. 2017).

What has changed in more recent years is the increased focus by parents, their supporters, and some service providers on the human rights of disabled people. This is supported by the UN CRPD (United Nations 2006) as well as domestic legislation in the case of Iceland, which reflects these developments (Lög um þjónustu við fatlað fólk með langvarandi stuðningsþarfir 38/2018 2018). However, these developments do not appear to have engendered much of a change in how child protection systems, at least in Iceland, view and treat parents with disabilities. If anything, some recent cases illustrate that the child protection system and their lawyers have responded to these human rights developments with a shift in tactics to ensure the desired outcome. The purpose of this contribution is to report on the findings of a child protection case concerning a demand for the permanent removal of a child from a parent with IDD that wound its way through the Icelandic legal system as an example with which to demonstrate this shift.

The first component of the argument Is referred to herein as the aura of professionalism. One persistent criticism of Icelandic child protection throughout our work (see e.g., Stefánsdóttir et al. 2022a, 2022b; Rice and Sigurjónsdóttir 2022; Sigurjónsdóttir and Rice 2017, 2020) is that support provided to parents with IDD is not extensive; sometimes inappropriate, if not outright harmful; and often provided by those who have little or no training in supporting these parents and understanding their needs. In the case under consideration here, it appeared at first glance that child protection tried to provide extensive and professional support to the parent. However, after an in-depth analysis of the evidence, it became clear that this particular case was indeed worked poorly, yet, in a manner of speaking, wrapped in a package of professionalism. The tentative hypothesis is that when confronted with the growing awareness of disability rights and scrutiny of their work, child protection cannot continue its usual working methods when dealing with parents with disabilities. Enveloping the working methods in this case with the "aura of professionalism" with the inclusion of professionals who claim to have expertise in this area, child protection signals an awareness that greater legal protections for disabled parents mandate additional support measures. The analysis of the materials suggests that there is still not extensive awareness of what such support entails or should entail, despite what appears to be some willingness on the system's part to try. However, the end result is that the inclusion of this aura of professionalism sends an even stronger message to

the courts that all means have truly been tried and that the case was worked adequately enough to justify custody deprivation.

The second component of this argument is referred to herein as *alleged disabilities*. Earlier research (see e.g., Sigurjónsdóttir and Rice 2017) has demonstrated how the impairments of the parent under investigation were often exaggerated, even intensified, over the duration of the case. The argument was that this was intended to solidify the basis for custody deprivation by making the parent's impairments appear even more severe than they were. Yet, up until the case under consideration here, there had not been a prior example we are aware of in Iceland where the parent's impairments and disability status were minimized, questioned, or even denied outright. The hypothesis presented herein is that when confronted with counterarguments against custody deprivation framed in terms of disability rights, and in explicit reference to the CRPD and Icelandic legislation inspired by the Convention, the denial of disability status or the minimization of the impairment type or degree becomes yet another tactic to circumvent progress in disability rights and, thus, continues to lead to denial of the right to parent on the part of people with disabilities. What follows is a brief discussion which makes our approach to disability clear. Finally, we conclude with a discussion of the Icelandic legal context pertaining to relevant disability legislation, and some details of the case under investigation herein, to provide further context.

### 1.1. Models of Disability: The Human Rights Approach

For the purpose of analysis, we are adopting a human rights perspective on disability, although in our data we can see elements of other perspectives. What has come to be seen as the social model of disability, largely originating in the UK, posits that individual impairments become disabling when one encounters barriers that are societal in origin. This model challenged the dominant medical approach to disability, which focused on the body and its impairments as the causes of disablement. One task for activists was to shift the attention to social arrangements in order to define, eliminate, or reduce these barriers. The process of doing so has also been argued to have contributed to the empowerment of disabled people, as the social model has also become a "vehicle for developing a collective disability consciousness" (Oliver 2013, p. 1024). A cultural approach to disability is also of importance here, because through this perspective, the focus is largely placed upon how disability is represented in cultural forms (Waldschmidt 2018). This includes the prejudice and stigmas that are an integral part of these child protection cases in our experience, and which are often commented upon by parents. The Nordic relational model of disability focuses on the interplay between the environment and the individual. As described by Ytterhus et al. (2015): "Disability is understood as resulting from complex interactions between the individual and the socio-cultural, physical, political and institutional aspects of the environment" (p. 21). Elements of the social model and relational approach also underpin the ideology of the CRPD and the human rights approach to disability. O'Mahony and Quinn (2017) argue that the human rights approach to disability builds upon the social model's focus on social change by calling for specific transformations to be made and positioning disabled people as rights holders. Barriers do need to be removed, but this approach formulates barriers as human rights violations. Degener (2016) argues that the CRPD is "the first human rights instrument which acknowledges that all disabled persons are right holders and that impairment may not be used as a justification for denial or restrictions of human rights," and which "recognizes that disability is a social construct which is created when impairment interacts with societal barriers" (Degener 2016, p. 1). Our argument is not that all instances of child custody deprivation concerning disabled parents are human rights violations per se, and that disability-specific human rights violations can and do happen within the child protection system. This case is one such example, despite the enhanced attention paid to the rights of disabled people in recent Icelandic legislation.

*1.2. The Icelandic Legal Context*

The United Nations Convention on the Rights of Persons with Disabilities (UN CRPD) entered into force in 2008. The Convention places disability within a human rights framework, with an emphasis upon the agency and decision-making capacities of people with disabilities, in contrast with the older medical or charitable understandings of disability which situated disabled people as objects of care (Brennan et al. 2016; Löve et al. 2017; Quinn 2009). Of particular relevance for the context of parenting with a disability is Article 23—respect for home and the family, section 4 of which states: "In no case shall a child be separated from parents on the basis of a disability of either the child or one or both of the parents" (UN CRPD). However, this is permissible "when competent authorities subject to judicial review determine, in accordance with applicable law and procedures, that such separation is necessary for the best interests of the child (UN CRPD)." The key issue here lies with the notion of "competence" and whether the parent at the centre of the case we have analysed was provided adequate support and assessed properly. Iceland signed the CRPD on 30 March 2007 and later ratified it on 23 September 2016. After ratification, the Icelandic government intensified it, aiming to replace older domestic disability legislation with new laws which reflect the language and aims of the CRPD. This has proven to be a long and complex process, initiated pre-ratification in 2010 (Löve et al. 2017), but it has continued for years afterwards. Of particular importance here is the Act regarding services to disabled people with long-term support needs, 38/2018 (Lög um þjónustu við fatlað fólk með langvarandi stuðningsþarfir 38/2018 2018). Chapter 1, Article 2 of this law arguably reflects elements of the social as well as Nordic relational models of disability, as it posits that disability arises as the result of the interaction between the impairments of individuals and barriers of various kinds, such as environmental and attitudinal obstacles, that hinder full and effective participation in society on an equal footing with others. Chapter 3, article 8 of this legislation defines the support services in Iceland available to disabled people, arguing that these supports are essential to the goal of an inclusive society, especially mentioning (Article 8, sub-section 5) the needs of disabled parents in the care and upbringing of their children. However, due to the dual nature of the Icelandic legal system, this process has been slow and uneven. As Gjecaj et al. (2023) write: "the Convention needs to be transposed into Icelandic law to formally gain the status of national law" (p. 5). As of the time of writing, this process has yet to be completed. Some of the legal tools to protect and support parents with disabilities are in place in Iceland. For example, the 2011 Act on the Protection of the Rights of Disabled Persons (PRDP Act No. 88/2011) provides regional disability rights protection officers (RPOs), who are tasked to safeguard the rights of disabled people and assist them. Though some of these legal tools exist, in our experience, and in the case under consideration here, there is a gap between the ideals of the CRPD and these laws and practices as they unfold on the ground in child protection work. We have found little evidence of knowledge of the CRPD within the Icelandic child protection system. We have repeatedly seen little reference to the CRPD in the cases we have analysed, and this lack of knowledge has been confirmed for us by other researchers, legal professionals, and parents who are well-informed about their rights. The case under consideration here is an example of how the process unfolds, and how it can lead to custody deprivation even when disability rights-based arguments are made by the parents, their lawyers, and the disability rights protection officers that adequate measures were not put into place.

*1.3. The Facts of the Case*

The case concerns the demand for permanent custody deprivation pertaining to the child of a parent with IDD. The Icelandic Supreme Court declined to hear the case and supported the decision of the lower Icelandic Court of Appeal (*Landsréttur*), which determined that custody deprivation was warranted. As of the time of writing, the father and his lawyers have submitted this case to the European Court of Human Rights. This case demonstrates that the long-standing concerns raised by scholars pertaining to the treatment of parents with IDD by the child protection system remain in place, at least in

Iceland. This case is one of many which demonstrates this, but it was chosen as an exemplar not only due to its particularly egregious nature, but also to how it reveals the more recent tactics under discussion here.

The parent at the centre of the case works full-time, and the pregnancy of the couple was not planned. The couple split apart, and it became a child protection case with a focus on the primary care provider at the time, the mother. The mother spent a period of time at a child protection training home (*vistheimili*) where parenting skills could be observed and evaluated and training could be provided. Shortly afterwards she withdrew from the training home and gave up custody of the infant. The other parent, the father, who later became the focus of this custody deprivation case, indicated his desire to raise the child. Due to concerns about the father's impairments and lack of experience in childcare, child protection decided that the original plan would continue on in the training home, with the father replacing the mother. This plan continued for a period of three months, after which he received a positive evaluation from the staff and it was decided that the father and his child would leave the training home to a shared accommodation with the grandparent, the father's mother. The general consensus appeared to be, as evidenced within the case data, that the father could take care of the child with extensive support. He was to receive follow-up visits four times a week in the home from staff from the training home, and twice per week for six weeks from a member of hired company described as being able to provide "specialised service due to disability." This would be complemented by a support family provided through social services from whom the father could periodically receive some respite from the pressures of parenting while adjusting to his new reality. He also received assistance and assessment from an organization that specialises in therapy for families with young children and parent–child connection issues. On paper, this support scheme appeared to be well-conceived and ideal to suit his needs. In the narrative of the case for custody deprivation later presented to the courts, it appeared as though extensive and professional disability-specific support had been tried and had failed. However, with a careful examination of the data, a counter-narrative becomes apparent, which holds that what was really presented to the courts was a failure that arose from a number of ill-conceived and haphazard measures cloaked in the aura of professionalism, not a failure that arose after the implementation of meaningful and practical support that the parent needed to be successful in the parenting role.

## 2. Methods

The primary dataset for this study consisted of selected court case documents provided to us with permission from the father or shown to us during interviews with him. Face-to-face interviews, email exchanges, and telephone conversations with the father and the grandparent about their experiences were also used, in addition to some discussions with the parent's lawyers and advocates. Research of this nature is arguably enhanced if the analysis of texts is complemented by interviews with parents, family, and supporters. One reason is that the voices of those most affected by custody deprivation cases, such as the parents, children, and extended family, are often the voices that are the least prominent in these materials. The views and perspectives of the child protection system and the professionals they employed are included by way of the substantial documents they produced over the duration of this case, rendering the need to interview them directly somewhat moot. Furthermore, the basic quality of the reports entered into evidence can be quite poor at times, and the information needs to be cross-checked. Within this case, and others we have seen, more often than expected the children at the centre of the cases in these documents are mis-gendered or mis-named; mistakes are made about ages, names, and roles of individuals; various details, dates, and timelines are incorrect; and so forth. This has not gone unnoticed by parents and grandparents, whom we have heard mocking these errors, and speculations sometimes arise that those compiling the documents perhaps "cut-and-paste" texts from other cases to use as templates, only changing key details, in order to save time. Interviews are, as such, also necessary to fill in gaps and to make

corrections about the sometimes confusing details recorded in these documents. Some of the case documents were entered into the data analysis software Atlas.ti, and the authors collectively read through the material and discussed its significance while simultaneously coding the material through the use of in vivo codes for specific terms and phrases. The coding process was informed by the principles of Grounded Theory (Strauss 1987; Corbin and Strauss 2014; Charmaz 2014). The analysis was constructed out of the themes extracted from the data, which later led to the development of a set of axial codes once the focus grew sharper. By way of example, in vivo codes included elements such as various disability- or impairment-related terminology. From there, we analysed the context in which these terms were used and how they were written about. With the use of the software, we were quickly able to collate the adjacent contextual material in order to create higher-level analytical themes, such as the themes that form the basis of our analysis here. Further questions and gaps in knowledge were addressed by regular meetings held with the parent at the centre of this case, and continued well after legal appeals were exhausted. The analytical process also contained some comparative elements, such as consideration of the effects (or not) of recent legislation in Iceland compared with cases from earlier research projects. The analytical approach is also influenced by critical disability studies, with a particular emphasis on the human rights approach to disability, as discussed in Section 1.1. The particular case under consideration here proved to be a rather unique for us in that the disability concept and its meaning became an explicit point of contention among the concerned parties at one critical juncture in the case. "Disability," from a psychological perspective, primarily functions as a biomedical diagnostic category. In this case, from the point of view of child protection workers, disability primarily represented a risk factor pertaining to child abuse or neglect. From the father's point of view, the disability concept engaged with his self-identity and understandings of himself as a disabled person. For lawyers and disability rights protection officers, the focus of the disability concept was that of human rights and legal protections. Our approach takes note of all of these dimensions of disability, but due to our training in disability studies (first author) and anthropology (second author), we are also drawn to considering the broader contextual factors in which disability/impairment is embedded, such as, among others, interpersonal relations, family dynamics, material resources, organizational power, and issues of culture, such as prejudice, stigma, and disability representation.

*Ethical Considerations*

The research for this contribution was supported by the University of Iceland's research fund. This project was submitted for commentary to the Ethics Committee of the University of Iceland, which determined that the study does not contravene the University's Code of Ethics and which had no reason to oppose the study. All names and identifying details have been omitted or altered in order to protect the privacy of the parents and the extended family, as well as various professionals connected to the case. No access to the primary data was provided to anyone beyond the two authors, and this material was stored on password-protected laptops. No direct content from any of these documents was reproduced in this contribution, aside from what has been made available to the public in redacted formats. The only specific details that are included are those deemed necessary for the coherence of the narrative, but deemed not to compromise the identities of the parents, family, or professionals involved. The origin of this particular contribution lies in the request from the father's lawyers to turn our analysis into an academic article that could be referenced if required. Another was the desire of the father to have his story told, and he played an active role in discussions about the material and events about which we should write. There is a long tradition within disability studies of (non-disabled) scholars acting as allies of disabled people to help further the goals and aid in the struggles of disabled people and their families, although this history of alliance is also riddled with examples of less-than-helpful benevolence (Soldatic and Johnson 2019, p. 2). It is also important to bear in mind that parents with IDD who are subjected to child protection interventions face enormous

imbalances in power; the child protection system and the medical and legal professions are stacked against marginalised parents, oftentimes with only small numbers of family and friends in their corner. Whatever criticisms of bias that may be raised are, from our point of view, offset by our attempts, however small they may be, to address these power imbalances through acts of research-based advocacy. Ultimately, our goal is to work toward supporting a more just and fair child protection system for families headed by parents with disabilities, but such work can never be completed if these injustices and poor working methods are neither recognised nor critiqued.

## 3. Findings

### 3.1. The Aura of Professionalism

The interpretation of the support that was provided to the parent as having merely an "aura of professionalism" was inspired by the work of an Icelandic scholar in gender studies, who argued that gender relations in Iceland are presented within the "aura of gender equality", which advances a positive messaging, but, in fact, obfuscates a more complex and less positive reality (Pétursdóttir 2009). The argument presented herein follows a similar logic, in that what was presented to the courts was positive messaging about the scale, scope, and quality of support that the parent was provided, but lacked a careful and critical consideration of the provided support measures, which would be necessary in order to understand whether they were well-designed and implemented. Without this necessary step, it is not possible to adequately assess the parenting ability of the father or the environment in which he would perform this role.

### 3.1.1. At Home and Alone

Shortly after the father and child left the training home to reside with his mother, they found that the grandmother had already left the country for a family emergency. This fact did not result in a change to the support plans, even though a psychologist noted that the parent should not be sent to an empty home unless proper support was in place. According to the first parenting assessment, as evidenced in the District Court (Héraðsdómur) ruling and reiterated in the finding by the Icelandic Court of Appeal (Landsréttur), the psychologist argued that the father would not be able to provide his daughter with "acceptable upbringing conditions" and supervision unless such support was in place. In the District Court ruling, this statement from the first parenting assessment was used as evidence of the father's inability to parent, as evidenced by the psychologist. However, no mention was made that child protection must have thought otherwise, as the caution of the psychologist was ignored and the father was indeed sent to an empty home. This could be argued in a number of ways which could have favoured the father's case: that child protection's actions suggest that they considered that the father had a greater parenting capacity than that indicated by the psychologist; that the court opted to emphasise the evidence that cast the father in a negative light; or that the "specialised" support provided by child protection was not extensive and the father managed in spite of this. During this period, there was also little documentary evidence of in-home visits by the staff from the training home, though the father informed us that some visits were made by the staff during this time, but they were not particularly useful. He articulated the issues confronting him in the parenting role, but it does not appear that this information was acted upon. During this time, we noted several instances in which the father stated he was lonely and had some difficulties with domestic duties such as meal preparation, but there was not much indication of skill-based training being provided. He confirmed in an interview that these visits were of little help. It must be remembered that this coincided with the height of the COVID-19 pandemic in Iceland, which negatively impacted general social interaction along with the provision of in-home support due to staff being ill or in isolation. An organization that specialises in therapy for families with young children wrote a report during this time which spoke positively of the parent and his attachment with the child. However, practical in-home support did not appear to be provided during

this time. Later in the case documentation, there is no mention of this organization, except a comment made by the father some five months later that they had not heard from the psychologist from this organization for a "long time."

Perhaps of greater concern was the company that was hired to provide in-home support to the parent, initially twice per week for six weeks. The testimony from an employee of this company played a pivotal role in the evidence that would later be used in court to justify custody deprivation. From the documents analysed, there is little evidence that the staff from the training home and this company coordinated their efforts or shared information. From what little information could be gleaned from this company's web and social media presence at the time, there is no evidence of competence in child protection matters. Furthermore, this company hired a contract worker with a professional clinical background and some training in supporting people with IDD, but with no evidence of education or training in supporting parents with IDD specifically in the parenting role. From the case files, there is little on display, which speaks to a methodology regarding assessing or defining parenting strengths and weaknesses, planning for targeted interventions which take into consideration the parent's impairments or the home and social environment, or measuring outcomes within a realistic and appropriate timeframe. What was provided initially was support for 12 h over the course of 37 days, which amounted to visits to the parent in the home twice a week for one-hour intervals. There was a second period of support that amounted to nine visits over six weeks, with some two to three hours per visit. There were two further short periods of support after this, but they occurred after the decision was made to move forward with the process of custody deprivation and were of dubious utility.

What was reported from all of this was a series of observations about the parent, the interactions with the child, the home environment, and some small excursions outside of the home, such as on walks and visits to a local park. Some comments and tips made to the father about matters that a general social worker could offer, not at all in keeping with the claimed expertise associated with this organization. The case evidence provided by this worker suggested that her role was, first and foremost, one of surveillance and information gathering on behalf of child protection to provide practical support. Further, some of the observations about the home were puzzling, such as an observation that a raised flower planter posed a potential danger to the child, which was actually raised in court. Little of what was referred to as support appeared to be useful to the father, especially when they clearly defined some of their needs, such as with cooking, cleaning, and, at the time, assistance with infant care, all of which could be provided by social workers from the municipal social services with experience in supporting disabled parents. The lack of practical help from these visits was confirmed for us by the father during interviews.

The parent was also let down in other ways. He repeatedly stated in the case data that he was alone, stressed, and tired, and needed a support family on some weekends when the child would be at home. What was provided was a support family during the middle of the week. In a document referring to a treatment plan at the start of the process, it is noted that the child would spend four days a month for a period of six months with a support family. The father's desire was for this to occur on weekends as the child was in daycare when he was at work, but this request appeared to be ignored for reasons that are not stated or else were included in documentation to which we do not have access. The inappropriateness of this was even noted by the worker from the support company as well as the daycare provider, both of whom found this to be a poor decision. The material suggests that there were differences of opinion and even clashing views about the parent and his parenting ability among those tasked to assess this as well as the measures that needed to be implemented. This is not unusual in custody deprivation cases. In a Canadian study (Aunos and Pacheco 2021), it was argued that the disparity between the inherently negative view of parenting with a disability on the part of child welfare workers and the more strengths-based approach of services that specialise in supporting parents with IDD has to do with their differing roles, education, and training. However,

the criticism of the decision here was notable, as was the lack of response to changing the arrangement. Further, when comparing and contrasting the arguments presented before the courts during different stages of the case, these kinds of internal disagreements, as well as positive assessments of the father, tend to be minimized or vanish altogether in favour of negative examples in order to justify the case for custody deprivation. Overall, the support that was provided was not extensive; it did not match the parent's stated needs nor was it particularly specialised, despite child protection's repeated claims to the contrary. What is difficult to determine from the case data was whether child protection genuinely believed that the support provided was adequate, or if they were not aware of what was required in order to support the parent properly. Neither scenario is positive, and the courts appeared to be willing to accept child protection's narrative regarding this matter that all means had been tried.

This process of, for lack of a better word, "cherry-picking" of the evidence was notable in terms of how the parenting assessments performed by psychologists in this case were used to shape the view of the parent presented to the court. Legal systems tend to prefer their experts in this matter to be psychologists, and Iceland is no exception, as psychologists are routinely called upon to perform this role. Questions have been raised in the literature about whether or not this is the best practice, particularly if the assessor does not have significant experience in interviewing and assessing people with IDD. This demand is particularly troubling given the tendency of psychologists to infer a strong relationship between parenting capacity and IDD (Aunos and Pacheco 2021). The other issue is that credentialism appears to feed into this aura of professionalism. An assessment by a psychologist or other such professional will carry greater weight in evidence presented to the court than what could be important counter-evidence provided by those who may have greater first-hand experience with the parent and family, but may lack similar professional stature. Given the weight routinely attributed to parenting assessments, the initial views of the parent were primarily shaped by the parenting assessments that were conducted in a rather short period of time by a number of different psychologists. The findings, or, more accurately, elements of them, did figure prominently in the later evidence provided to the courts. However, it was surprising, given knowledge of the outcome of the case, how positive these initial assessments were, which highlighted many of the father's positive characteristics along with the expected statements about the parent's weaknesses and need for support. As the narrative of the case developed, it became apparent that the backbone of the evidence supporting the argument for custody deprivation was instead constructed from the negative reports produced by the support worker from the company hired by child protection. This support person played the dual role of providing support as well as engaging in surveillance and information gathering, but appeared to place, or was required to place, an emphasis upon the latter. This observation was further supported when it was learned that the hired support worker from this company was asked to collect information about the father and the relationship with his child from the daycare provider, although it would be expected that this should be the duty of a formal child protection worker, if not the case manager. A repeated reference was also made in the reports articulating some concerns about the parent's impairments. These were made by a public health nurse who, as far as could be seen from the documents, did not know the father or the case well and only spent very limited time with him and the child during a routine check-up. It was initially not clear why so much emphasis was placed upon the nurse's commentaries given their obvious deficiencies as evidence—that is, until her negative views of the parent fit the later narrative in support of custody deprivation and her status as a nurse appeared to help to strengthen the aura of professionalism around the evidence that child protection presented to the courts.

### 3.1.2. A "Bonding Disorder"

There is evidence in the parenting assessment process of both negative and positive findings. The positive findings, in particular, contrasted with the narrative being con-

structed by child protection during the custody deprivation phase. For example, one major theme in the argument set forth is that there was a disorder in the bond (Is. *tengslaröskun*) between the father and the child, and that the child's development was at risk because of a lack of stimulation. It was initially unclear from where this "bond disruption" argument originated. The initial parenting assessments expressed no concern about this at all, and a support organization which specialises in therapy for families with young children as well as connection disorders wrote a positive report on the bond between the parent and the child as they transitioned out of the training home. A partial answer to this question appeared in a later assessment, where the psychologist described the process whereby the parent and the child were permitted to leave child protection's supervised housing unit. The father, unhappy with the existing parenting assessments, demanded that his parenting ability to be assessed with regard to the possible support resources and services to which he was entitled according to 38/2018 law and CRPD. The psychologist claimed that disability was his speciality and initially rejected input on the case from a disability studies scholar, as both the father and his rights protection officer requested. The psychologist later claimed that his assessment included the view of a disability studies scholar, but we later learned this was a person who was not familiar with the case details nor specialised in disability and parenting issues. The support provided in the home was tasked to an organization which claims to specialise in supporting people with disabilities. This specific worker also has a background as a clinician, all of which appears impressive and adds to this aura of professionalism. It appeared that the origins of the narrative of a bonding disorder came from the reports of this worker, who also argued that the child showed behaviours linked to the effects of serious neglect. However, this claim was not corroborated by the evidence from the daycare provider, who spent more waking time with the child than anyone else during the same time period. However, the daycare provider's evidence was not clinical in nature, but rather gained from observations grounded in experience and without the credentials typically respected by the courts. This claim about a "bonding disorder" was re-articulated in the third parenting assessment, in which it is stated that "the child had showed serious symptoms of attachment disorder (tengslaröskunar)" when the child was removed from her father. However, when this assessment was conducted, the child had already been removed and was in temporary care. We surmised that this information was based upon the views of the support worker when the child was with her father, but when rearticulated by the psychologist in the parenting assessment, who solidified and amplified the evidentiary weight of the claims made by the support worker. We later learned from the father that when the worker from the support agency was challenged in court on the claims regarding a bonding disorder between parent and child, the worker backpedalled and claimed there was no definitive evidence of a connection disorder, but rather "possible signs." Nonetheless, this distinction did not appear to matter. Under this aura of professionalism, statements about "concerns" or possible "signs" of parental neglect become infused with such a level of gravitas when applied to disabled parents that they become accepted as evidence—they become interpreted as affirmative statements about neglect. Once this narrative is in place, and the outcome of custody termination is the sought-after goal, this aura also serves to deflect criticism on the basis of professional competency from those who seek to challenge their assertions.

The argument being made here does not claim that psychology or the other clinical sciences have no competency or value in the area—quite the opposite. Such assessments need to be carried out by qualified professionals. The claim is that the evidence needs to be reviewed more critically than it is, and that there appears to be such an unwillingness to interrogate the evidence associated with specific credentials that it precludes this level of examination. A degree in a clinical science does not mean there is any specific training or experience involved in working with parents with IDD on matters of parenting and support. In response to statements found in the parenting assessments, such as "[the father] has difficulties reading in to [his daughter's] needs and feelings and therefore it is a danger that her attachment will not develop normally", the father argued that the psychologist did

not answer the question about possible support resources and services to which he could be entitled according to 38/2018 law that could help to increase his custody ability. He further explained to us that he felt that child protection did not fully research what support services he could have utilized as a disabled person, that what was provided was not enough, and that the assessment did not take this into consideration. The ruling from the court was that the psychological assessment was "not deficient" (ekki slíkum annmörkum háð) and, therefore, there was no reason for further assessment. Supporting disabled people as an organization's expressed competency can mean many things. Little more than enveloping claims of risk of neglect or signs of a disruption in the parenting bond within the aura of professionalism is needed to tip the scales against disabled parents.

### 3.2. Alleged Disabilities

Disability is often a common reason for both child protection interventions and custody deprivation itself, and (low) measured IQ plays an oversized role in these cases. For example, in a study of American appellate cases concerning the termination of parenting rights that involved IDD and measured IQ, in 86% percent of these cases, low measured IQ of the parents was presented to the court as a barrier to parenting (Callow et al. 2016). This resulted in the decision to uphold the original order to terminate custody in 81% of these cases. The authors concluded that "the courts consistently considered parental IQ, rarely reviewed evaluation methods and results and frequently made statements that reflected a view of parental IQ as static, fixed and necessarily undermining of parenting capacity and ability to learn" (ibid. p. 560). Questions about the over-reliance of the courts on IQ levels in custody termination cases were raised long before this, as is evident in Galliher's 1973 criticism (Galliher 1973). As a result, a low measured IQ or a diagnosed impairment becomes linked to claims about sub-standard parenting and, in turn, plays an important role in how evidence is interpreted and how outcomes are assessed.

In the case under consideration here, from a clinical point of view, there was some disagreement among the psychologists about the scale and extent of the father's diagnosis of ID, but there was general agreement about the other impairments. Nevertheless, the parent's impairments were initially discussed in predictable ways: statements which attested that he had difficulty learning new things; all parenting shortcomings being explained by the impairment; and ultimately, the decision that he would not be permitted to parent without a significant level of support. From the point of view of child protection (and confirmed by the courts), the tailored support provided to the parent was claimed to be extensive, and the parent was not able to apply it in practice. As discussed earlier, this support was of dubious quality and efficacy considering the stated needs of the parent. When he lost custody of his child at the district court level, the judges explained that their decision was based largely upon the parent's impairments, and that due to this, he lacked competence in the parenting role despite the support that was provided. Disability was used as a disqualifier from the parenting role, and from this line of reasoning, custody termination became the only imagined outcome.

After the case was appealed and moved to the appellate court, there was a notable and quite surprising shift in tactics regarding how child protection's lawyer dealt with the issues of disability. This was also reflected in a parenting assessment in which the psychologist referred to the father's "alleged disability", which, to the best of our collective knowledge, is an argument that had not been raised before in a custody deprivation case pertaining to parents with disabilities in Iceland. The psychologist went on to explain that some of the parent's diagnoses were not up for debate, but questioned whether the assessed IQ was low enough to constitute a level of impairment that would denote intellectual disability. Further, he explained that certain forms of autism can be very disabling, but the parent in question appeared to have a mild form, citing his ability to make eye contact and the good communication skills he possessed. The fact that the parent worked full-time was noted numerous times in the case data as well. The contradictory images of the parent's mannerisms and level of impairment in the data were stark at times, and

sometimes it appeared as though entirely different people were being discussed in these documents. This development necessitated a detailed examination of how this alleged disability process unfolded.

For the first year and a half of this process, most of those concerned with the case accepted the parent as someone with multiple diagnoses who required support in the parenting role. The first appearance of the argument that the parent may not be a disabled person was raised in a later psychological assessment, where the "alleged disabilities" comment was made. What had changed at this point in the case was that the father had sought help from one of the disability rights protection officers in Iceland (Is. *réttindagæslumenn fatlaðs fólks*), who injected the possibility of disability rights violations into the proceedings. Prior to this time, the case was worked on the part of child protection from the understanding of the parent's impairments. This approach appeared to be a reasonable way in which to work the case. However, as the parent's lawyers began to make the argument that the father was not provided the appropriate support to which he was entitled under Icelandic law as a disabled person, another occurrence of the "alleged disabilities" narrative appeared. Rather perplexingly, in one statement written by the lawyer for child protection, it was argued that the selection of support measures took into consideration the father's impairments, and then proceeded to detail how his impairments were taken into account at all stages of the case. However, in the same document, it was later argued that the parent was not disabled according to the legal definition of disability in Iceland. The relevant legislation in Iceland is the Act on services to disabled people with long-term support needs 38/2018 (Lög um þjónustu við fatlað fólk með langvarandi stuðningsþarfir 38/2018 2018). Child protection consistently argued that the parent's impairments hindered his ability in the parenting role. Chapter 3, Article 8 of this legislation defines the support services in Iceland available to disabled people, arguing that these supports are essential to the goal of an inclusive society, especially mentioning (Article 8, sub-section 5) the needs of disabled parents in the care and upbringing of their children. In total, this presented a strong argument that the parent had impairments, did not stand on equal footing with other members of society in some regards (here, in the parenting role), and required support to meet these needs. As discussed earlier, the father argued that he needed to be provided the assistance and training required to parent as a disabled person and to which he was entitled under the law. However, he repeatedly argued that what was provided was not sufficient. The father informed us that the questions the judges asked indicated that they had already made up their minds up about the case, and in his view only asked questions that confirmed their opinions about his custody ability being limited rather than focusing on questions about the support provided. Based upon the questions asked, the father also disputed that the judges knew what kinds of support was available for disabled parents in general, let alone for his specific needs.

The shift in tactics appeared to be a direct response to the arguments of the father's lawyers at the appellate level, which framed the case as a disability rights violation. Prior to this time, there was little disagreement about the parent's status as a disabled person, and child protection understandably worked that case based on this understanding, as they should. The father's impairments were diagnosed in childhood and youth, and for them he received medication and was directed toward suitable educational and training programmes, all of which were well-documented in the data. The arguments which denied the parent's disability status also revealed a lack of understanding of the disability concept in general, as well as how it is defined in various ways in Icelandic law and service provision. One example is that the parent did not undergo an evaluation according to the Supports Intensity Scale (SIS), and somehow, failing to do so called into question his disability status. The SIS assessment was developed by the American Association on Intellectual and Developmental Disabilities to assess the support needs of people with developmental disabilities who are over the age of 16 to live in the community (AAIDD 2022). This assessment had nothing to do with parenting and did not assess competence in this area, nor was it relevant for the assessment of the particular support needs required

in this case. Unfortunately, the SIS has been widely adopted in Iceland as a measure of disability and used in ways for which it was not intended. Next it was pointed out that the parent never went through the disability pension assessment (Is. *örorkumat*). In Iceland, this assessment first and foremost concerns reduced working capacity and focuses on assessing an individual's entitlement for a pension due to reduced working capacity following illness or disability (Tryggingastofnun 2022). Neither of these applied in this case, as the parent's impairments did not interfere with work capacity in his particular job and, thus, there was no need for such an assessment, nor would it provide much useful information about parenting. Finally, a statement was made that the parent "seems to have good social relations with friends and family." In a passage that was solely concerned with questioning disability status, the obvious interpretation of this stateme It is that it suggests that one of the defining features of Iisablement is that one does "not" have good social relations with friends and family, which is both absurd and insulting to disabled people at large. Months later, when discussing the case, the father mocked this tactic, sarcastically commenting "Well, I guess I am not a disabled person because I have good relations with friends and family." The appellate court appeared to view the "alleged disabilities" argument with some scepticism, but nevertheless upheld the district court's ruling, which supported custody termination.

## 4. Discussion

While the challenges to the parent's disability status through the "alleged disabilities" tactic were dismissed by the father and his supporters, this final argument that asserted he was not disabled because it appeared that he had good social relations troubled the parent in particular. The implication of this statement was clear to all those concerned and was raised in multiple interviews. Custody deprivation proceedings are painful and traumatic for the parents, the children, and extended family and friends. It is also emotionally difficult for parents to read what is written about them and the arguments that are made to justify custody termination. The lack of knowledge of disabilities and the lives of disabled people on the part of the system was put on full display by this tactic. The tactic which questioned disability status served only to attempt to avoid the protections afforded to disabled people via the CPRD, and the domestic legislation in Iceland was influenced by it to ensure that the demand for custody deprivation would be agreed upon by the courts. The primary injustice in this case is that custody deprivation could have been avoided if appropriate support measures that were put in place were provided by those trained and experienced in supporting parents with IDD in parenting roles. What was offered was a mix of poorly planned and haphazard supports that emphasised intelligence gathering at the expense of implementing appropriate support measures. As such, the process was wrapped in the "aura of professionalism" with the knowledge that support framed as providing disability-specific support would strengthen the case for custody deprivation if needed in an enhanced disability rights environment. This was complemented by the alleged disabilities tactic as the case moved to the appellate level and the arguments shifted to that of disability rights violations. The additional injustice is that this tactic revealed how little the child protection system still understands disability and the needs of disabled parents, and that it resorted to an argument which appeared to assume that a defining feature of disablement is that disabled people all live lives in isolation, bereft of family and friends. If that is indeed the dominant view of disability by child protection, it is not surprising why there is such a determined effort to remove children from disabled parents once such cases come to their attention. Along with the more recent tactics discussed herein, this case reveals how little has changed since the pre-CRPD years and how much work ahead there is still to do.

**Author Contributions:** Writing—original draft, H.B.S. and J.G.R. All authors have read and agreed to the published version of the manuscript.

**Funding:** This research was funded by Rannsóknasjóður Háskóla Íslands (University of Iceland Research Fund) for the project *Disability, immigration and multigeneration: intersecting factors in child protection cases* (2020–2022).

**Institutional Review Board Statement:** The project *Disability, immigration and multigeneration: intersecting factors in child protection* cases (2020–2022) was submitted for commentary to the Ethics Committee of the University of Iceland, which determined that the study does not contravene the University's Code of Ethics and had no reason to oppose the study (Vísindasiðanefnd Háskóla Íslands—7.4.2020).

**Informed Consent Statement:** Informed consent was obtained from all subjects involved in the study.

**Data Availability Statement:** The data are not publicly available due to ethical and privacy issues.

**Conflicts of Interest:** The authors declare no conflict of interest.

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
