# Peer review of "“Alleged Disabilities”: The Evolving Tactics of Child Protection in a Disability Rights Environment"

_laws, 2022_

Round 1

Reviewer 1 Report

The manuscript presents a case study in Iceland that reflects the lack of implementation of the UNCRPD in daily practice of children protection and deprivation of custody. It provides insight and, perhaps more importantly, details of how this could happen in a European country.

I would suggest following adjustments, if possible, for the benefits of reader:

1. Adopt more substitles in Section 3 on findings. It would make it easier for readers to understand the 'story' and analysis.

2. Include more (translated though) direct quotes from courts' ruling and/or the interview which would significantly improve the vividness of the case.

Author Response

Thank you for your work and support for this submission. According to your review the criteria you were asked for received a positive response.

Reviewer #1 suggested two ways to improve the manuscript:

1.) Add more subtitles in Section 3.

We agreed those sections are long; we have added some sub-sections to help with the readability – specifically 3.1.1 At home alone, and 3.1.2 A ‘bonding disorder.’

2.) More direct quotes from the court documents. We agree 100% that this would improve the vividness of the piece. However, when the submission was originally drafted we were working primarily with unpublished court documents, marked ‘confidential,’ that the father allowed us to view, but we felt that ethically, and legally, it would be unwise to use direct quotes. After the piece was submitted, a redacted version of the case from the appellate court was released. At various points in the text we included some direct, and translated, quotes from the case. However, in the process of doing so, in combination with reviewer #2’s request for more theory and legal analysis, the revised version started to push 10.000 words and we felt it wise to be cautious with regard to how much to add.  

Reviewer 2 Report

This article has a contribution to make to the literature by introducing an international audience to a key issue and the approach of the Icelandic authorities and what this reveals about the effectiveness and application of The United Nations Convention on the Rights of Persons with Disabilities (CRPD). However to communicate this to the reader your article needed to give a little more context to the legal framework concerning the rights of Icelanders with disabilities, how it has developed and how it interesects with supranational frameworks and stakeholders. The reader would also have benefitted by having the facts of the case to be analysed brought towards the start of the article, as the structure is currently hard for readers to follow.

In terms of the analysis, the discussion of Art. 23 of the CRPD needed greater nuance and application, because in as much as it prohibits the removal of children from family setting on the basis of disability alone, there is provision provided that the correct procedures are followed. You need to explain in greater depth to the reader why this is not the case within the current ruling.

In terms of your framework, you also need to make your view of disability more explicit to the reader. For example you talk about the impact of professionalism in decision making, but do not refer to the Medical Model, nor do you engage with the Human Rights Model sufficiently in your discussion of the CRPD.

Author Response

Thank you for your work on this submission. A number of changes were asked for that in our view helped to improve the piece. However, we are not clear on why it was marked “Extensive editing of English language and style required.” No examples were provided. One of the co-authors is a native English speaker who has been publishing in English language academic forums since the mid-2000s. While no text is perfect and there are always those who can improve any piece of writing, we remain unclear as to what the issues are that would require extensive language editing. One possibility is that we made the conscious decision to not name the parent as a ‘father.’ We were, at the time, working with texts that were private and not open to the public, and opted to not make the parent’s gender clear for ethical reasons. This resulted in, perhaps, some clunky language. However, since these texts were made public, in a reacted form, the parent is named as a father, and we have since made this change. We have also paid additional attention to proofing matters in this round and hopefully this will satisfy this criticism. If not, we request some examples of the text from the reviewer of language that requires extensive editing.

1.) The reviewer commented that more context of the legal framework is required. Thank you for this observation and we agree fully – that is indeed lacking. We wrote a passage that will hopefully satisfy this criticism. See inserted text on pp. 3-4.

2.) The reviewer commented that the facts of the case need to be brought forward at the start of the article as the structure is difficult to follow. We agree that this is indeed a problem. In response, we have moved the facts of the case from section 2.2 (original version p.7) into the newly revised introduction. This section is now referred to as 1.3 and can be found on pp. 4-5.

3.) We agree that the provision within Article 23 that children may still be removed under certain circumstances was not made clear. We have added some text about this in the (new) section 1.2. We have also added details at various points in the revised version to make the case stronger that the correct procedures were not followed. While the focus of the article is on newer tactics used by child protection to skirt around the protections within the CRPD and those transposed to Icelandic law, we agree that we need to strengthen the basic case that proper procedures were not followed in a more general way. Added text and details can be found in pages 7 and page 10. We contend that the original material, on page 8, also provides a good deal of information to argue that correct procedures were not followed and that the specialist support provided was neither specialist nor effective as support. Some editing was done on page 12 to strengthen the argument as well.

4.) The reviewer commented that our specific stance on disability needed to be made explicit. We agree with this, and have include a passage on this matter. We thank the reviewer for this, as we were able to also make the argument stronger because of its inclusion. See the inserted new text on p.3, as well as some elements in the (new) section 1.2.

Other changes:

1.) During the original draft, we had only private court documents provided by the father to work with. For ethical reasons, we decided to conceal his gender. This perhaps lead to some awkward wording that may be at root of reviewer #2’s comments about the need for language editing, as well as some confusion. Once the public documents were made available online, it is clear that the parent in question is a father, so we have now made this change in various places to reflect this. All such changes are marked in track changes. General proofreading was also done with an aim to raise the quality of the text where possible. All changes have been marked with track changes.

Round 2

Reviewer 2 Report

The resubmission adequately addresses the issues identified in the original submission. The written expression is much clearer and the structure is easier to follow. The contextual information added also gives greater clarity to the overall arguments and broadens the interest to an international readership.